# TOWARDS MULTIMODAL UNDERSTANDING, REASONING, AND TOOL USAGE ACROSS VISION, SPEECH, AND AUDIO IN LONG VIDEOS

## ABSTRACT

Long-form, multimodal video understanding requires models to integrate vision, speech, and ambient audio while reasoning coherently over extended contexts. However, existing benchmarks often emphasize either long temporal contexts or rich multimodal content, but rarely both. Moreover, they are typically restricted to multiple-choice evaluations and a single accuracy metric, offering limited insight into where models succeed or fail. To address these gaps, we introduce **STARBench**, a diagnostic benchmark designed for long-form, multimodal video understanding. STARBench features open-ended, intent-driven questions that reflect how humans naturally engage with video content. It supports single- and multi-turn dialogues, encompassing multimodal reasoning and agentic tool-use tasks across rich video, audio, and speech contexts. Each question includes a reference answer and a rubric with graded criteria, enabling interpretable and traceable evaluation. Importantly, STARBench is generated via a scalable, human-validated pipeline, ensuring reproducibility and coverage. Complementing the benchmark, we propose **STARAgent**, an agentic system for analyzing long videos using pre-processing, search, and refinement tools. Evaluating state-of-the-art closed- and open-source MLLMs on STARBench reveals substantial limitations: the top-performing Gemini-2.5-Flash reaches only 52.95%, while open-source models remain below 25%. STARAgent, leveraging structured reasoning over long videos, achieves 44.66%, highlighting the challenge of complex, real-world video understanding. By combining breadth, interpretability, and reproducibility, STARBench provides a practical foundation for benchmarking and improving MLLMs on long-form, multimodal video tasks. All code, including the agentic pipeline, and datasets will be released publicly.

## 1 INTRODUCTION

Large language models (LLMs) have gradually expanded from text-only reasoning to handling increasingly diverse input modalities. Early efforts centered on textual understanding, followed by extensions to vision (Bai et al., 2025; Yang et al., 2025; Wang et al., 2025) and audio (Google, 2024; Xu et al., 2025). More recently, omni-modal models have sought to unify text, vision, and audio within a single framework (Xu et al., 2025; AI et al., 2025), enabling broader forms of multimodal reasoning. In parallel, benchmarks have been developed to reflect this trajectory: from text-only tasks, to vision-language datasets, and now to multimodal audio-visual-language evaluations.

Within this landscape, video stands out as both natural and challenging. It requires models to jointly process visual, speech, and ambient audio signals, while maintaining temporal coherence across extended sequences. Prior benchmarks on video understanding have typically focused on short, manually trimmed clips (li et al., 2022; Li et al., 2023; Patraucean et al., 2023; Ning et al., 2023; Yang et al., 2022; Mangalam et al., 2023; Chen et al., 2024; Liu et al., 2024; Geng et al., 2023). With the rapid improvement of multimodal large language models (MLLMs), however, such short-form clips are no longer sufficient to reveal their strengths and limitations. As a result, recent work has turned to long-form videos, sometimes lasting an hour or more. Yet despite advances in model architectures and training corpora (Chen et al., 2023b;a; Lee et al., 2021), evaluations of long-form video understanding remain fragmented, shallow, and difficult to interpret.

Existing benchmarks often trade off between temporal length and modality coverage. Some handle long videos but ignore key modalities such as audio or speech (Wang et al., 2024b; Ataallah et al., 2025; Zhang et al., 2025), while others preserve multi-modality but focus on short clips or narrow tasks like video captioning or temporal grounding (Fu et al., 2024a; Geng et al., 2023; 2025; Li et al., 2025). Most benchmarks rely on multiple-choice questions, and even those with open-ended answers typically reduce evaluation to a single score. This oversimplification hides model failures in perception, cross-modal integration, and reasoning (Mangalam et al., 2023; Wang et al., 2024a; Zhang et al., 2023), limiting diagnostic insights and opportunities for improvement.

To address these limitations in existing benchmarks, we present **STARBench**, a new benchmark for long-form multi-modal video understanding that goes beyond typical vision-centric benchmarks by explicitly incorporating Speech, Tools, Audio, and complex Reasoning in long video contexts. Rather than an incremental extension of prior work, STARBench rethinks the benchmark paradigm itself. Our contributions are fivefold:

**(i) Holistic multimodal integration.** STARBench aligns audio, speech, and vision into a temporally consistent representation, capturing cross-modal interactions such as spoken references to off-screen events or sounds that help explain what is happening visually. By contrast, earlier long-video benchmarks often leave out raw audio or only include speech as written transcripts (Wang et al., 2024b; Ataallah et al., 2025).

**(ii) Intent-driven questioning.** STARBench generates scenario-driven questions that reflect the different ways people watch videos, such as looking for facts, understanding causes, planning actions, or using tools. Questions appear in single- or multi-turn Q&A sequences that mimic natural interactions. They cover general understanding, reasoning, and tasks where models must actively gather information using specialized tools rather than relying only on memorized knowledge. In contrast, previous benchmarks focus on narrow, fixed tasks (Mangalam et al., 2023; Lin et al., 2025; Geng et al., 2025).

**(iii) Diagnostic, interpretable scoring.** Instead of relying on a single score, STARBench pairs each question with a weighted rubric that breaks success into clear criteria, such as factual completeness, temporal localization, modality grounding, and tool use. Independent evaluation of these rubrics provides fine-grained diagnostics, highlighting both strengths and weaknesses, and enables partial credit when models demonstrate partial understanding.

**(iv) Practical scale with human rigor.** STARBench scales to hour-level videos through a combination of automated, chunk-based dense caption, Q&A generation, and systematic human validation. This approach ensures both wide coverage and reliability, and the full pipeline will be released to support reproducibility and community-driven expansion.

**(v) Agentic method.** We introduce **STARAgent**, a fully integrated agentic pipeline designed to reason over long videos using pre-processing, search, and refinement tools. STARAgent is able to seamlessly integrate information across time, leverage multimodal inputs, and execute tool-based strategies, setting a new standard for practical, real-world video understanding. By pairing STARBench with STARAgent, we provide both a diagnostic benchmark and an actionable pipeline, demonstrating how MLLMs can move beyond static evaluation to tackle complex, temporally extended tasks.

More details on the construction of STARBench, including video statistics, task categories, and our human validation pipeline, are provided in Sect. 3. We evaluate STARAgent alongside both closed-source and open-source omni-modal MLLMs, such as Gemini-2.5-Flash (Google, 2024) and Qwen2.5-Omni (Xu et al., 2025). STARAgent surpasses open-source baselines and achieves performance comparable to Gemini on agentic tasks. Nevertheless, all approaches continue to struggle with long-form video reasoning, particularly on hour-long videos, where overall performance remains far below expectations. To advance progress in this area, we also conduct extensive ablation studies that provide insights and guidance for improving multimodal long-form video understanding.

## 2 RELATED WORK

### 2.1 BENCHMARKS FOR LONG-FORM VIDEO UNDERSTANDING

To better probe the capabilities and limitations of current MLLMs, recent studies have shifted towards long-form videos, including hour-level durations. Several benchmarks have been proposed for

Table 1: **Comparison between STARBench and existing video understanding benchmarks.** STARBench balances multiple modalities (Visual, Audio, and Speech) and supports intent-driven and tool-augmented Q&A, multi-turn interactions, and rubric-based explainable scoring.

| Benchmarks | Visual (V) | Audio (A) | Speech (S) | Intent-Driven Q&A | Multi-Turn Q&A | Open-Ended Q&A | Hierarchical Rubrics | Tool-Usage |
|---|---|---|---|---|---|---|---|---|
| MV-Bench (Li et al., 2024b) | ✓ | × | × | × | × | × | × | × |
| EgoSchema (Mangalam et al., 2023) | ✓ | × | × | × | × | × | × | × |
| LongVideoBench (Wu et al., 2024) | ✓ | × | × | × | × | × | × | × |
| Moviechat (Song et al., 2023) | ✓ | × | × | × | × | ✓ | × | × |
| MLVU (Zhou et al., 2024) | ✓ | × | × | × | × | ✓ | × | × |
| LvBench (Zhang et al., 2025) | ✓ | × | × | ✓ | × | × | × | × |
| VideoMarathon (Lin et al., 2025) | ✓ | × | × | × | × | ✓ | × | × |
| LVBench (Wang et al., 2024b) | ✓ | × | × | × | × | × | × | × |
| UnAV-100 (Geng et al., 2023) | ✓ | ✓ | × | × | × | × | × | × |
| InfiniBench (Ataallah et al., 2025) | ✓ | × | ✓ | × | × | ✓ | ✓ | × |
| Video-MME (Fu et al., 2024a) | ✓ | ✓ | ✓ | × | × | × | × | × |
| LongVALE (Geng et al., 2025) | ✓ | ✓ | ✓ | × | × | ✓ | × | × |
| TriSense-2M (Li et al., 2025) | ✓ | ✓ | ✓ | × | × | ✓ | × | × |
| **STARBench (ours)** | ✓ | ✓ | ✓ | ✓ | ✓ | ✓ | ✓ | ✓ |

this purpose. For example, LVBench (Wang et al., 2024b) contains 103 curated high-quality videos with 1,549 manually generated question-answer pairs, targeting six core capabilities such as summarization and temporal grounding. However, LVBench primarily emphasizes visual modalities (i.e., video frames) and excludes audio signals. InfiniBench (Ataallah et al., 2025) and LvBench (Zhang et al., 2025) incorporate long-duration video and subtitle contexts, but audio understanding remains underexplored. More recent efforts have sought to establish omni-modal benchmarks for evaluating MLLMs. Videos are typically sampled from large-scale corpora such as VAST-27M (Chen et al., 2023b), VALOR (Chen et al., 2023a), and ACAV-100M (Lee et al., 2021), which provide rich audio-visual annotations. Specifically, LongVALE (Geng et al., 2025) and TriSense-2M (Li et al., 2025) incorporate audio, visual, and speech modalities. However, LongVALE only features videos averaging around 4 minutes, while TriSense-2M extends to approximately 15 minutes. Moreover, although these datasets broaden multimodal coverage, their Q&A tasks remain largely confined to segment captioning and temporal retrieval, limiting evaluation of advanced MLLM reasoning and understanding. Table 1 presents an overall comparison between STARBench and existing benchmarks. In summary, STARBench achieves a better balance between video duration and multimodal understanding, and also stands out for its unique question and rubric design.

## 2.2 UNIQUE CHARACTERISTICS OF STARBENCH

**Question Design in Video Q&A.** Existing benchmarks adopt human-authored or template-driven queries. EgoSchema (Mangalam et al., 2023) emphasizes long-horizon reasoning but centers on duration rather than user intent. VideoMarathon (Lin et al., 2025) and ALLVB (Tan et al., 2025) rely on predefined task categories or prompt taxonomies, while GAIA (Mialon et al., 2023) and GTA (Wang et al., 2024a) focus on real-world grounding but remain bounded by fixed templates. LongVALE (Geng et al., 2025) and TriSense (Li et al., 2025) focus on the temporal video grounding and segment caption. These designs risk constraining model behavior to narrow expectations. STARBench addresses this limitation with scenario-driven, intent-aware questioning that covers single- and multi-turn interactions.

**Tool-Augmented Video Understanding.** While tool-augmented language models have been studied (Schick et al., 2023; Qin et al., 2023), their extension to multimodal video understanding remains largely unexplored. STARBench enables evaluation of this capability by requiring models to actively gather information via tools in video Q&A tasks.

**Evaluation Methodologies.** Most prior benchmarks rely on coarse-grained metrics, such as accuracy or exact-match scores, offering limited diagnostic insight. MoVQA (Zhang et al., 2023) and GTA (Wang et al., 2024a) propose more fine-grained metrics like cosine similarity or step-level evaluation, but interpretability remains limited. STARBench differs by incorporating weighted, rubric-based scoring for more granular analysis of model performance across multiple criteria.

## 3 CONSTRUCTING STARBENCH

Our STARBench is designed to evaluate the multimodal understanding capabilities of MLLMs in long-form videos, encompassing vision, audio, and speech. To this end, we develop a five-stage pipeline: **multimodal caption generation, question design, answer generation, hierarchical rubric generation, and human validation**. As shown in Fig. 1, this pipeline transforms raw

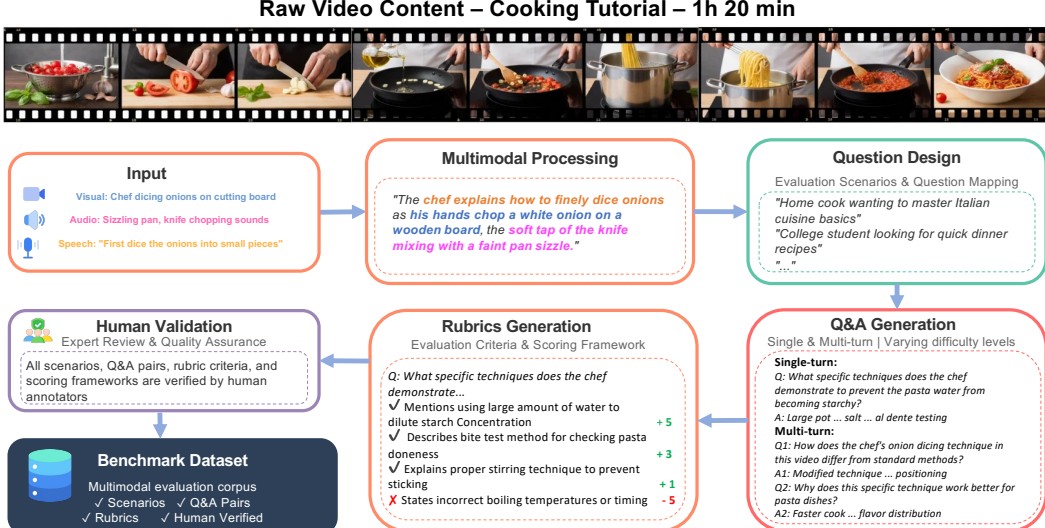

Figure 1: **Construction pipeline of STARBench.** The pipeline begins with raw video data where speech, visuals, and audio cues are extracted. These are passed into multimodal processing to generate segment-wise aligned and fused metadata. Only the distilled information flows to question design, where scenarios and question types are mapped, followed by the generation of questions and conversational answers. Next, verifiable rubrics are created to evaluate correctness and difficulty. Finally, the core dataset, comprising Q&A pairs and tailored evaluation rubrics, is manually reviewed and corrected by human validators, ensuring a clean, reliable benchmark.

multi-source videos into structured evaluation samples, where the questions are derived from diverse user-intent scenarios, and the rubrics-based evaluation provides interpretable and traceable model scoring. We elaborate on each stage in the following subsections.

### 3.1 MULTIMODAL CAPTION GENERATION

**Video Collection.** We collect long videos from Video-MME (Fu et al., 2024a), sampling a total of 92 videos. In addition, we manually curate 65 videos from YouTube to support the design of questions involving more complex agentic tool invocation. Overall, STARBench contains 157 videos with an average length of 44.7 minutes, totaling 117.46 hours. Each video comes with a high-quality audio track, which is used for speech and audio analysis.

**Modality-specific Captioning.** To efficiently process long videos, we split each video into segments based on speech activity, treating silent intervals or gaps between speech as natural segment boundaries. *Speech* content is then transcribed into time-stamped captions using Whisper-large-v3 (Radford et al., 2023). For *visual* content, representative frames from each segment are passed to the vision-language model Qwen-2.5-VL-32B (Bai et al., 2025) to generate dense, segment-level scene descriptions. *Audio* events such as music, applause, and environmental sounds are detected using Audio-Flamingo-3 (Ghosh et al., 2025). This segmentation strategy allows us to capture rich, modality-specific information while preserving temporal granularity, even in segments with little or no speech.

**Cross-modal Summarization.** The captions from all three modalities are integrated using Qwen3-30B-A3B-2507-Instruct (Yang et al., 2025) to produce coherent, high-level narratives. This model reconciles potential conflicts across modalities, abstracts complex interactions (e.g., "the presenter gestures toward a diagram while explaining a concept"), and aligns multimodal captions temporally.

### 3.2 QUESTION DESIGN

**Scenario Determination.** Given the raw video and its corresponding multimodal captions, we aim to generate questions that are both video-relevant and diverse. Prior benchmarks often directly prompt an LLM or MLLM to produce questions, but such approaches make it difficult to ensure diversity and controllability in the generated results. In contrast, we introduce an intermediate step of *scenario determination* before formal question creation. A scenario represents a plausible viewing context in which different individuals may raise distinct questions, reflecting varied perspectives

Table 2: **Task taxonomy of STARBench for comprehensive video understanding evaluation.**

| Task Category | # Tasks | Description |
|---|---|---|
| **Core Perception** | 4 | Fundamental perceptual understanding encompassing entity recognition, event understanding, temporal understanding, and audio comprehension. These tasks evaluate the models ability to perceive and interpret visual and auditory information across time. |
| **Information Tasks** | 4 | Tasks focused on extracting and organizing information, including information retrieval, summarization, instruction extraction, and sentiment analysis. They assess the models ability to gather, condense, and interpret semantic content from videos and text. |
| **Multimodal Tasks** | 4 | Tasks requiring integration and alignment across multiple modalities, including multimodal synthesis, cross-modal verification, and audio-visual alignment. These evaluate the ability to reason and generate insights by connecting visual, auditory, and textual streams. |
| **Reasoning Tasks** | 4 | Higher-order cognitive tasks including causal reasoning, quantitative reasoning, compositional reasoning, and comparative analysis. These measure the models ability to infer, calculate, and reason about complex scenarios. |
| **Agentic Tasks** | 16 | Tool-augmented reasoning tasks leveraging perceptual tools (e.g., transcribe speech, detect faces, track objects), computational tools (e.g., calculator, execute code), and retrieval tools (e.g., cross-modal search, web search, memory tool). These tasks assess active information gathering and problem-solving capabilities. |

on the same video. For example, in a smartphone review video, a prospective buyer may ask about practical aspects such as battery life or app compatibility, while a tech enthusiast might focus on advanced features or camera comparisons. Scenario generation thus captures contextual intent, aligning subsequent tasks with both perceptual and reasoning challenges. To implement this, we employ Qwen3-30B-A3B-2507-Instruct (Yang et al., 2025), which analyzes multimodal video metadata, identifies salient entities, actions, and temporal relations, and produces up to five diverse scenarios per video. These scenarios are designed to cover a broad spectrum of viewer intents. The relevant prompt is provided in the Appendix.

**Task Assignment.** For a comprehensive evaluation of MLLMs, it is essential to balance the questions generated for the determined scenarios. To this end, we construct a task taxonomy that systematically organizes question types. As shown in Table 2, the taxonomy comprises five major categories: *Core Perception*, *Information*, *Multimodal*, *Reasoning*, and *Agentic* tasks. Each category further decomposes into multiple subtasks. In total, our STARBench encompasses **32** distinct tasks, providing broad and systematic coverage across perceptual, cognitive, and reasoning dimensions.

**Question Generation.** We design a systematic procedure to probe genuine multimodal video understanding. For each scenario, we prompt Qwen3-30B-A3B-2507-Instruct to generate two query types: (i) *single-turn* questions that are concise and self-contained (e.g., "What object is the chef holding in the first minute?"), and (ii) *multi-turn* conversations simulating dialogues with follow-up inquiries (e.g., a student asking several questions about an instructors demonstration). We control the question difficulty using a five-level scale: Levels 1–2 for simple recall/recognition, Level 3 for moderate reasoning, and Levels 4–5 for advanced temporal, causal, or contextual inference. To emphasize challenging reasoning, no more than 15% of questions are Levels 1–2, 25–30% are Level 3, 35–40% are Level 4, and 25–30% are Level 5. All questions are explicitly grounded in video metadata, referencing observable speech, visual frames, or audio cues.

### 3.3 Answer Generation

STARBench adopts the open-ended question-answering manner, which better aligns flexible conversation capability of MLLMs. Given the multimodal captions and generated questions, the Qwen3-30B-A3B-2507-Instruct is prompted to generate corresponding answers. We design several principles to ensure high-quality answer generation. First, the simple questions get brief replies, while complex ones receive more detailed explanations. The answers never invent or assume details not supported by the video metadata. Second, answers should integrate multiple modalities where relevant, linking actions, dialogue, and sounds to provide coherent explanations. Third, responses remain grounded and concise, avoiding over-analysis, technical jargon, or speculation about internal states unless explicitly observable. If a question cannot be answered from the available metadata, the system indicates this rather than providing uncertain or fabricated information.

### 3.4 Hierarchical Rubrics Generation

A key distinction of STARBench is its use of unique, verifiable rubrics for each Q&A pair.

**Criterion-based Scoring.** Each rubric consists of multiple criteria, designed via LLM prompting, that target specific facts or events from the video and can be verified by humans or LLM judges. For agentic tasks, the criteria assess query decomposition, tool invocation, parameter extraction, and integration of outputs (see Appendix A.2). The rubrics emphasize factual correctness over phrasing, focusing on entities, actions, numbers, and events. Unlike direct numeric scoring, which is often inconsistent, we query whether each criterion is satisfied, then compute the score automatically, leveraging LLMs strength in text understanding while avoiding instability in numeric judgments (Fu et al., 2024b).

**Priority Levels.** Criteria are weighted on a 10-point scale: high-priority (7 10) facts are essential, medium-priority (4 6) capture supporting details, and low-priority (1 3) add contextual enrichment. Factual errors incur negative weights. Final scores are calculated as the weighted sum of satisfied (+1) or violated (1) criteria. This system allows partial credit, rewarding models that capture the most critical facts while offering bonuses for completeness. Beyond grading accuracy, it provides transparency by linking scores to explicit criteria, enabling interpretable and fine-grained evaluation of reasoning quality.

### 3.5 Human Validation

To ensure the reliability of STARBench, we conducted a rigorous human validation process after automatic filtering. We recruited ten trained annotators with backgrounds in linguistics and computer vision, each familiar with multimodal annotation tasks. Before starting, all annotators underwent a structured training program, including (i) tutorials on the task taxonomy and rubric definitions, (ii) calibration sessions with sample videos, and (iii) group discussions to resolve ambiguities. This training ensured consistent interpretation of multimodal cues and standardized annotation practices.

**Validation Workflow.** Each video-question-answer triple was independently reviewed by two annotators. They verified that (i) the question is clearly phrased and grounded in observable video content, (ii) the provided answer is factually correct and complete with respect to the video metadata, and (iii) the evaluation rubrics faithfully capture the necessary facts and events. Inconsistencies or disagreements were flagged for adjudication by a senior validator, who made the final decision. To further safeguard quality, a random 10% subset was cross-checked by a third annotator, providing an additional layer of reliability monitoring.

**Revision and Removal.** Samples were removed if the video was corrupted, too ambiguous, or lacked sufficient multimodal cues. Questions were revised when phrasing was unclear or introduced unintended bias. Answers were corrected if they omitted essential details, introduced hallucinations, or used overly technical phrasing. Rubrics were adjusted to eliminate redundancy or to clarify verification criteria. This iterative refinement ensured that the final dataset balanced naturalness, clarity, and factual correctness.

## 4 STARAgent Pipeline

We introduce **STARAgent**, a **training-free agentic framework** (Fig. 2) that orchestrates a suite of specialized models and external tools for advanced multimodal understanding. At its core, STARAgent uses Qwen3-4B (Yang et al., 2025) as the orchestrator, which dynamically coordinates expert modules including Qwen2.5-VL-7B (Bai et al., 2025) for vision-language reasoning, Whisper-large-v3 (Radford et al., 2023) for refined speech transcription, Whisper-small for preprocessing, and Audio-Flamingo-3 (Ghosh et al., 2025) for audio understanding. STARAgent also leverages external tools such as activity detection, web search, and other task-specific APIs to enhance reasoning and data retrieval. The preprocessing module performs frame sampling via scene detection, background noise and non-speech audio analysis, SigLIP-based visual embedding extraction (Zhai et al., 2023), OCR, and lightweight speech transcription, while the refinement module handles deeper analysis with computationally intensive models for nuanced transcription, complex vision-language reasoning, and detailed audio interpretation.

During task execution, STARAgent's orchestrator adaptively decides which modules and external tools to invoke based on the query. Broad exploratory questions may trigger external search and preprocessing across modalities, while targeted queries on specific temporal segments can bypass search and directly use refinement modules. Sequential chaining is supported, such as preprocessing to identify candidate regions followed by refinement for detailed analysis. The orchestrator maintains contextual memory of prior outputs to avoid redundancy and ensure coherent multimodal

Figure 2: **STARAgent Pipeline.** The orchestrator agent (Qwen3-4B) receives a user query and video input, then calls the Preprocessor to extract multimodal signals, including Whisper-small speech transcription, scene-based frame sampling, SigLIP embeddings, OCR, and audio analysis. These features populate a vector database, which the Search tool queries to retrieve top-k relevant segments via semantic similarity. For deeper analysis, the orchestrator invokes Refiner tools such as Whisper-large-v3 for high-quality speech transcription, Audio-Flamingo-3 for detailed audio understanding, and a Video Refiner for dense caption generation. Beyond these core modules, STARAgent can access external tools including activity detection, web search, and other APIs to expand reasoning and retrieve additional context when needed. By flexibly sequencing preprocessing, search, refinement, and external tool calls, the orchestrator integrates multimodal evidence and auxiliary knowledge to generate a coherent final answer, demonstrating adaptive and agentic coordination across heterogeneous capabilities.

reasoning. By flexibly coordinating these capabilities, STARAgent demonstrates agentic behavior, dynamically selecting the optimal combination of models and tools for each query instead of following a rigid pipeline.

## 5 BENCHMARKING STARBENCH

### 5.1 EVALUATION PIPELINE

We evaluate candidate MLLMs on STARBench using a two-stage pipeline:

**Answer Generation**. The raw videos and corresponding questions are provided to the model for answer generation, which dynamically processes visual frames, audio, and speech (when supported). Notably, we do not sample a fixed number of video frames but allow each model to determine the quantity of video content it ingests. This design prevents stronger models, capable of handling longer sequences, from being artificially constrained, while also ensuring that models with more limited capacity are not overwhelmed by excessive input.

**Answer Evaluation**. The models' answers are compared against ground truth using the LLM-as-judge (Gu et al., 2024) framework. We employ Qwen3-14B (Yang et al., 2025), a strong open-source LLM for reproducibility. Specifically, the evaluation is conducted by separately checking each predefined rubric criterion, with the judge LLM constrained to only consider the relevant fact or event. This reduces ambiguity, improves reliability, and ensures verifiable scoring. After checking all criteria, we can obtain an overall score for a question-answer pair. Notably, for the multi-turn dialogue evaluation, to avoid error accumulation across multi-turns, we adopt an *ideal trajectory* setting. At each turn, the candidate model generates a response, which is stored for later evaluation. However, the running context for the next turn is updated with the *ideal response* rather than the model's output. This ensures that every turn is assessed independently under the same optimal history, isolating intrinsic response quality from conversational drift. Finally, we report the task-level scores by summarizing evaluation results of all samples according to the question task types.

Table 3: **Performance on STARBench (%).** **Gemini**: Gemini-2.5-Flash, **LLaVA-OV**: LLaVA-OneVision-Qwen2-7B-ov, **LLaVA-NV**: LLaVA-NeXT-Video-7B-hf, **Qwen-VL**: Qwen2.5-VL-7B-Instruct, **InternVL**: InternVL3.5-8B, **Qwen-Omni**: Qwen2.5-Omni-7B.

| | Gemini | LLaVA-OV | LLaVA-NV | Qwen-VL | InternVL | Qwen-Omni | STARAgent (Ours) |
|---|---|---|---|---|---|---|---|
| **Core Perception Tasks** | | | | | | | |
| Entity Recognition | 43.62 | 8.12 | 11.16 | 20.54 | 19.95 | 17.03 | 42.84 |
| Event Understanding | 41.84 | 6.66 | 9.60 | 14.21 | 14.47 | 13.95 | 35.41 |
| Temporal Understanding | 41.23 | 7.50 | 10.34 | 14.11 | 15.88 | 14.17 | 31.35 |
| Audio Understanding | 37.46 | 6.08 | 9.53 | 9.07 | 12.36 | 16.20 | 35.51 |
| Avg. | 41.04 | 7.09 | 10.16 | 14.48 | 15.66 | 15.34 | 36.28 |
| **Reasoning Tasks** | | | | | | | |
| Causal Reasoning | 68.41 | 9.01 | 14.43 | 24.76 | 23.98 | 23.73 | 54.26 |
| Quantitative Reasoning | 49.56 | 1.79 | 2.92 | 13.34 | 14.64 | 12.92 | 45.16 |
| Compositional Reasoning | 57.37 | 11.70 | 14.60 | 19.56 | 22.14 | 19.53 | 45.52 |
| Comparative Analysis | 71.24 | 9.73 | 13.61 | 20.72 | 20.05 | 17.93 | 52.87 |
| Avg. | 61.65 | 8.06 | 11.39 | 19.59 | 20.20 | 18.53 | 49.45 |
| **Information Tasks** | | | | | | | |
| Information Retrieval | 61.02 | 9.14 | 13.42 | 18.78 | 21.41 | 18.60 | 48.87 |
| Summarization | 58.86 | 12.92 | 18.22 | 18.61 | 22.72 | 19.16 | 60.17 |
| Instruction Extraction | 46.53 | 8.33 | 9.90 | 14.62 | 15.91 | 13.04 | 38.47 |
| Sentiment Analysis | 52.18 | 5.31 | 7.86 | 13.10 | 14.03 | 15.23 | 33.70 |
| Avg. | 54.65 | 8.92 | 12.35 | 16.28 | 18.52 | 16.51 | 45.30 |
| **Multimodal Tasks** | | | | | | | |
| Multimodal Synthesis | 55.38 | 8.92 | 11.34 | 19.14 | 18.99 | 16.59 | 44.15 |
| Cross Modal Verification | 50.45 | 4.60 | 10.91 | 10.81 | 12.31 | 11.21 | 37.79 |
| Audio Visual Alignment | 50.89 | 9.33 | 15.19 | 21.05 | 24.23 | 22.41 | 44.26 |
| Motion Analysis | 61.22 | 16.98 | 47.17 | 40.57 | 54.72 | 42.45 | 64.15 |
| Avg. | 54.49 | 9.96 | 13.82 | 22.89 | 27.56 | 23.17 | 47.59 |
| Overall | **52.95** | 8.51 | 13.76 | 18.39 | 20.49 | 18.39 | **44.66** |
| **Agentic Tasks** | 40.27 | - | - | - | - | - | 38.25 |

Table 4: **Performance on various video durations.**

| Duration | Gemini | LLaVA-OV | LLaVA-NV | Qwen-VL | InternVL | Qwen-Omni | **Ours** |
|---|---|---|---|---|---|---|---|
| 0-30 mins | **55.6** | **19.4** | **19.4** | **34.6** | **32.9** | **32.5** | **45.9** |
| 30-60 mins | 48.6 | 9.5 | 10.0 | 16.1 | 21.8 | 15.5 | 41.7 |
| >60 mins | 47.2 | 5.7 | 16.7 | 13.7 | 16.5 | 16.4 | 40.5 |

## 5.2 RESULTS AND ANALYSIS

Following the above evaluation pipeline, we assess the zero-shot performance of several representative closed-source and open-source models, spanning both omni-modal and vision-language paradigms. The models include Gemini 2.5 Flash (Google, 2024), Qwen2.5-Omni-7B (Xu et al., 2025), LLaVA-OneVision-7B (Li et al., 2024a), LLaVA-NeXT-Video-7B (Zhang et al., 2024), Qwen2.5-VL-7B (Bai et al., 2025), and InternVL3.5-8B (Wang et al., 2025).

**Main Results on STARBench.** As shown in Table 3, we report task-wise performance. For agentic tasks, we primarily compare against Gemini, given its strong multimodal tool-calling capabilities. We observe that omni-models generally achieve stronger overall performance than vision-language models (VLMs). Gemini-2.5-Flash achieves the highest overall score of 52.95% across four major task types, substantially outperforming all open-source models and demonstrating the advantage of jointly leveraging vision, audio/speech, and language modalities. Among open-source models, InternVL3.5-8B slightly surpasses the omni-model Qwen2.5-omni-7B, likely due to its larger parameter size and stronger vision capabilities. However, Qwen2.5-omni performs better on audio/speech-related tasks such as *Audio Understanding* and *Sentiment Analysis*. Except for InternVL3.5-8B, Qwen2.5-omni-7B shows a clear advantage over other VLMs. Nevertheless, the overall performance of open-source models remains below 25%, underscoring the current limitations of state-of-the-art MLLMs in multimodal understanding and reasoning, and highlighting the challenges posed by our benchmark. By contrast, our STARAgent achieves performance comparable to Gemini on both general and agentic tasks, while significantly surpassing open-source models. These results show the effectiveness of our method in retrieving useful multimodal information with relevant tool invocation.

Table 5: **Ablation results on modality used for question answering.**

| Modality | Gemini | LLaVA-OV | LLaVA-NV | Qwen-VL | InternVL | Qwen-Omni | Ours |
|---|---|---|---|---|---|---|---|
| A | 43.4 | **34.5** | **32.7** | **50.7** | **55.1** | **56.5** | 35.3 |
| S | **54.0** | 17.5 | 18.8 | 27.3 | 25.8 | 26.5 | **46.4** |
| V | 25.2 | 20.2 | 19.3 | 32.2 | 31.2 | 30.4 | 24.9 |
| A+V | 18.8 | 8.7 | 11.2 | 15.8 | 16.0 | 15.0 | 20.1 |
| S+V | 53.8 | 7.5 | 11.1 | 17.1 | 17.9 | 16.3 | 43.3 |
| A+S+V | 48.2 | 5.3 | 7.8 | 12.6 | 13.8 | 14.5 | 36.6 |

Table 6: **Performance on different question-answering types.**

| Type | Gemini | LLaVA-OV | LLaVA-NV | Qwen-VL | InternVL | Qwen-Omni | Ours |
|---|---|---|---|---|---|---|---|
| Single-turn | 51.0 | 5.1 | 9.2 | 14.3 | 15.9 | 15.2 | 41.4 |
| Multi-turn | **52.3** | **8.8** | **11.9** | **18.0** | **18.3** | **16.4** | **43.0** |

**Ablation on Video Duration.** Table 4 reports the results across different video durations. We divide the benchmark into three ranges: short (0-30 mins), medium (30-60 mins), and long (>60 mins). Overall, Gemini-2.5-Flash consistently achieves the highest performance across all durations, with our proposed STARAgent pipeline ranking second. Moreover, a clear trend emerges: most MLLMs perform best on shorter videos, and their performance degrades as the video length increases. For instance, Gemini-2.5-Flash drops notably from 55.6% on short videos to 47.2% on hour-long ones. These findings suggest that existing MLLMs still struggle with long-horizon multimodal understanding and reasoning. This highlights the urgent need for future research to design models with improved long-context handling, efficient memory mechanisms, and scalable multimodal reasoning capabilities.

**Ablation on Modality used for Q&A.** Our benchmark includes questions that involve either a single modality or multiple modalities. Table 5 reports the ablation results. Overall, models tend to perform better when the question only requires information from a single modality. For instance, our STARAgent achieves relatively strong accuracy on questions grounded solely in the speech modality (46.4%). In contrast, its performance drops as the number of required modalities increases (36.6% for 'A+S+V'). This suggests that analyzing and integrating information across different modalities remains a significant challenge: errors in individual modalities or conflicts among them can easily mislead the model and hinder accurate reasoning. These findings highlight the limitations of current MLLMs in multimodal understanding, particularly in long-video scenarios where multiple streams of information must be simultaneously aligned and interpreted.

**Ablation on Q&A Turns.** Our benchmark includes both single-turn and multi-turn question answering for each video. Table 6 presents the corresponding ablation study. Interestingly, the performance gap between the two settings is small. In fact, several models, including STARAgent, Qwen-Omni, and LLaVA variants, perform slightly better in multi-turn scenarios. For example, our agentic pipeline improves from 41.4% (single-turn) to 43.0% (multi-turn). This trend can be attributed to the construction of our benchmark: in multi-turn Q&A, later questions are often contextually related to earlier ones, and we provide the ideal correct answers for history turns to prevent error propagation. As a result, models can benefit from the additional context provided by previous turns, which may serve as helpful hints. Nevertheless, the improvements remain modest, suggesting that current MLLMs have not yet fully exploited dialogue history for reasoning and memory.

## 6 CONCLUSION

We present STARBench, a comprehensive diagnostic benchmark for evaluating MLLMs on long-form, multimodal video understanding, integrating vision, speech, and audio across hour-long contexts. Its open-ended, intent-driven questions and rubric-based evaluation provide fine-grained, interpretable diagnostics across perception, reasoning, and agentic tool-use tasks. Our experiments reveal substantial gaps in current state-of-the-art models, underscoring the persistent challenges of coherent multimodal reasoning over extended temporal contexts. By releasing both STARBench and STARAgent, our agentic pipeline for structured long-video analysis, along with a scalable, human-validated annotation framework, we enable reproducible research and provide a practical foundation for future model development. Collectively, these contributions aim to guide the next generation of MLLMs toward robust, real-world video understanding and advance progress on complex, multimodal reasoning tasks.

## 7 ETHICS STATEMENT

We use publicly available or permissioned long-form videos for research; we exclude sensitive data; and all annotations were performed by adult annotators who agreed to participate. We carefully filtered the videos for sensitive content including violence, explicit material, and personally identifiable information, implementing multiple review stages to ensure compliance with ethical guidelines. Any videos containing potentially harmful or inappropriate content were systematically removed from our dataset before annotation began.

## 8 REPRODUCIBILITY STATEMENT

We will release all benchmark code, agentic pipeline implementation, and evaluation protocols under CC-BY-NC-SA 4.0 license. Our release includes: (1) complete evaluation harness with fixed dependencies and random seeds, (2) curated test datasets with preprocessing scripts, (3) agentic pipeline configurations and prompting strategies, (4) all evaluation prompts and expert assessment rubrics (Appendix B), and (5) scripts to reproduce reported metrics. Hardware requirements and API specifications are fully documented.

Fair use of generative AI: AI tools were used only for code boilerplate and grammar refinement. All benchmark design, evaluation methodology, and numerical results were manually authored and verified. No proprietary data or undisclosed model outputs were used.

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

## A    DETAILS OF STARBENCH

### A.1    DATASET STATISTICS

The STARBench dataset is composed of 157 long-form videos, specifically curated to evaluate the long-context understanding capabilities of Multimodal Large Language Models (MLLMs). The benchmark dataset consists of 3338 Q&A samples. As illustrated in Fig. 3, the benchmark has a distinct focus on videos of substantial length to rigorously test model performance on extended temporal sequences. The video durations are heavily concentrated around a central peak, with a mean of 44.7 minutes and a median of 45.6 minutes. The close proximity of these two metrics indicates a relatively symmetric distribution. The histogram clearly shows that the vast majority of samples are clustered between 40 and 60 minutes, confirming that the benchmark is primarily built with near-hour-long videos. This characteristic makes STARBench a challenging testbed for evaluating multimodal integration, temporal reasoning, and information retention over significant durations. Fig. 4 visualizes detailed video category distribution and Fig. 5 provides some Q&A and rubric examples.

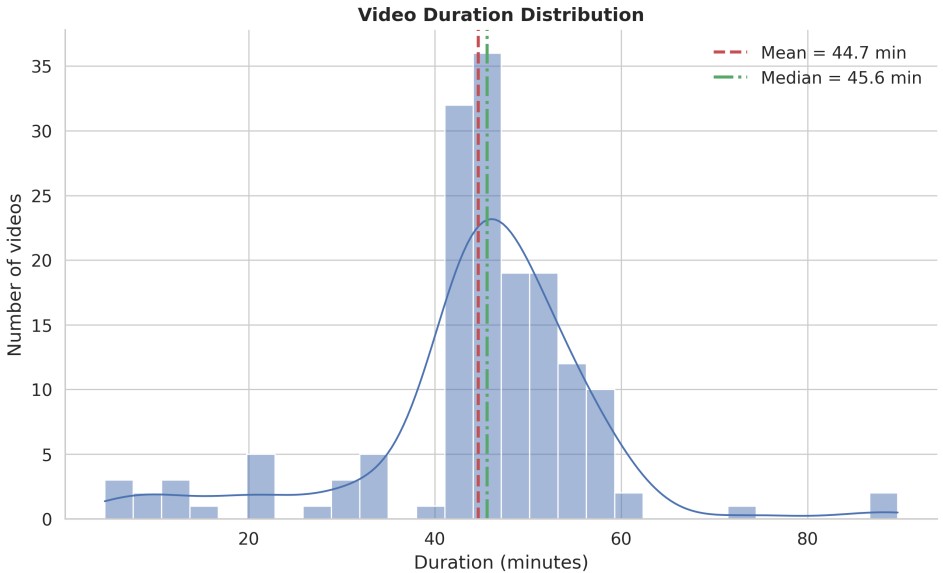

Figure 3: **Distribution of video durations (minutes)** in our validated sample set ($n = 157$).

### A.2    DETAILS OF THE AGENTIC TASKS

This section presents the complete inventory of tools available for agentic video understanding tasks in our benchmark. These tools enable multi-modal reasoning across visual, auditory, and textual modalities, supporting complex queries that require tool composition and sequential reasoning. The benchmark comprises 193 multi-turn conversation scenarios with 891 total tool invocations across 16 distinct tools. Each tool is designed to extract specific information from video content or perform computational operations on extracted data. Table 7 provides a comprehensive overview of each tool's functionality, parameters, and expected outputs.

The tools span multiple categories of functionality: (1) speech and audio processing for transcription and sound detection, (2) visual understanding for text extraction, object detection, and activity recognition, (3) translation services for multi-lingual content, (4) computational tools for calculations and code execution, and (5) information retrieval through web search and memory operations. This diverse tool set enables agents to solve complex real-world video understanding tasks that require coordinated use of multiple modalities and external knowledge sources.

Table 7: **Complete list of tools for agentic video understanding tasks.** Optional parameters are marked with (?).

| Tool Name | Parameters | Output | Description |
|---|---|---|---|
| `transcribe_speech` | `timestamp, language(?)` | `{transcript, timestamp}` | Extract spoken words from video segment with temporal alignment |
| `extract_scene_text` | `timestamp, language(?)` | `{texts[]}` | Extract visible text overlays, captions, or written content from frames |
| `count_objects` | `timestamp, object_query` | `{object, count}` | Count specific objects within video segment |
| `object_detection` | `timestamp, categories[]` | `{timestamp, object, bounding_box}` | Detect objects, logos, and brands with spatial localization |
| `extract_math_exp` | `timestamp, language(?)` | `{expressions[]}` | Detect and extract mathematical expressions from video frames |
| `calculator` | `expression` | `{result}` | Perform mathematical calculations on extracted values |
| `web_search` | `query, num_results(?)` | `{title, snippet, url(?)}` | Query external web sources for supplementary information |
| `detect_audio_events` | `timestamp` | `{description}` | Identify environmental sounds and audio events |
| `execute_code` | `code, language, inputs(?)` | `{stdout, stderr}` | Execute code snippets in sandboxed environment |
| `detect_activities` | `timestamp` | `{activities[]}` | Recognize human actions and activities in video |
| `summarize_segment` | `timestamp, text, length` | `{summary}` | Generate concise summaries of video segments |
| `detect_faces` | `timestamp` | `{faces[{name, role(?), confidence}]}` | Identify and recognize known individuals |
| `cross_modal_search` | `query, modalities, k` | `{timestamp, results}` | Find query occurrences across modalities (default k=5) |
| `memory_tool` | `func=[add\|delete\|search], data` | `{success\|fail, output}` | Store, retrieve, or delete intermediate results |
| `translate_speech` | `timestamp, from_lang, to_lang` | `{translated_transcript}` | Translate spoken content between languages |
| `translate_text` | `text, from_lang, to_lang` | `{translated_text}` | Translate written text between languages |

# B    PROMPTS FOR DATASET GENERATION PIPELINE

We provide the complete prompts used in our five-stage STARBench generation pipeline. Each stage takes structured outputs from the previous stage as input, ensuring consistency and preventing hallucination through careful information flow control. These prompts guide the automated generation of scenarios, questions, answers, and evaluation criteria from video metadata. Each prompt is carefully designed to ensure high-quality, grounded, and realistic benchmark data while avoiding hallucination and maintaining consistency across the pipeline stages.

## B.1    SCENARIO ANALYSIS AND TASK ASSIGNMENT PROMPT

The following prompt is used in Stage 2 of our pipeline to analyze video metadata and generate realistic viewing scenarios with associated evaluation tasks. This prompt emphasizes strict grounding in the video content to avoid hallucination while generating diverse perspectives for comprehensive evaluation.

---

**Scenario Analysis and Task Assignment Prompt**

```
You are analyzing video metadata to identify natural contexts and perspectives from
    which people would be curious about this video content. Think about realistic
    situations where someone would want to understand what happened in the video.

CRITICAL ANTI-HALLUCINATION REQUIREMENTS
========================================
- NEVER add details, objects, people, or events not explicitly mentioned in the video
    metadata
```

---

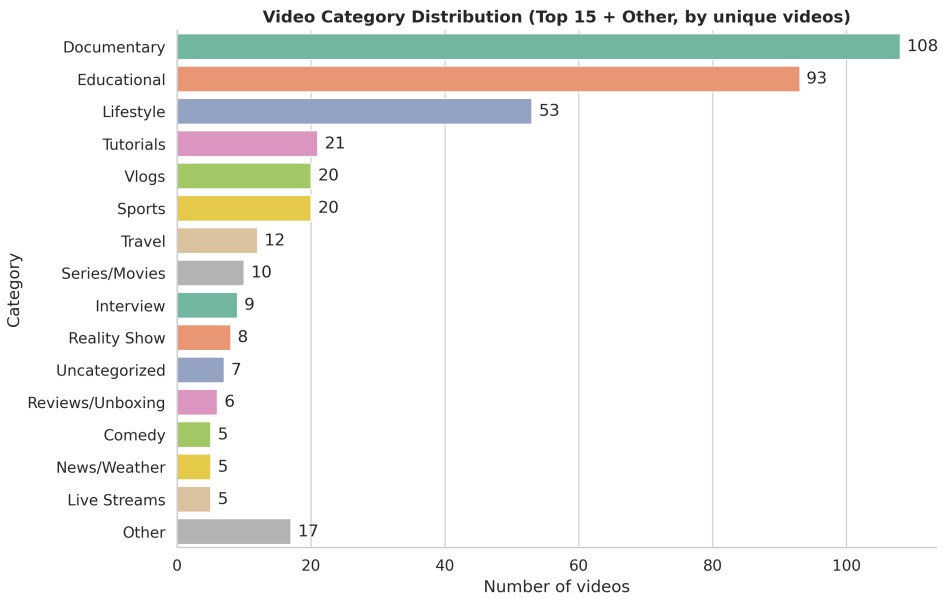

Figure 4: **Video category distribution.** Each video may belong to multiple categories.

```
- NEVER assume content based on typical expectations for video types
- NEVER infer activities, locations, or contexts beyond what is directly described
- ONLY reference what is explicitly documented in the provided metadata
- If metadata is sparse, create fewer but more grounded scenarios
- Every scenario element must be verifiable against the source metadata

Your Task

Your job is to:
1. **CAREFULLY** analyze the metadata to understand ONLY what is explicitly described
   in the video.
2. Generate diverse, realistic scenarios representing different natural human
   perspectives for viewing this content.
3. For each scenario, identify what kinds of questions a person in that context would
   naturally ask.
4. Map those natural curiosities to evaluation task categories (to ensure
   comprehensive testing).

Requirements
1. **Think like real people**: Consider realistic situations where someone would watch
   this video and be curious.
2. **Natural multimodal interest**: People naturally connect what they see, hear, and
   observe - scenarios should reflect this.
3. **Diverse perspectives**: Cover different reasons why people might watch (learning,
   entertainment, analysis, etc.).
4. **Distinct contexts**: Each scenario should represent a meaningfully different
   viewing situation.
5. **STRICT METADATA GROUNDING**: Every scenario must be completely supportable by the
   provided metadata.

Input:
Structured or free-form metadata describing the visual, auditory, and speech content
   of a video.
{video_metadata}

Output format (strict JSON):
{
  "scenarios": [
    {
      "scenario": "<concise scenario description>",
      "tasks": ["<task_1>", "<task_2>", "..."]
    }
  ]
}
```

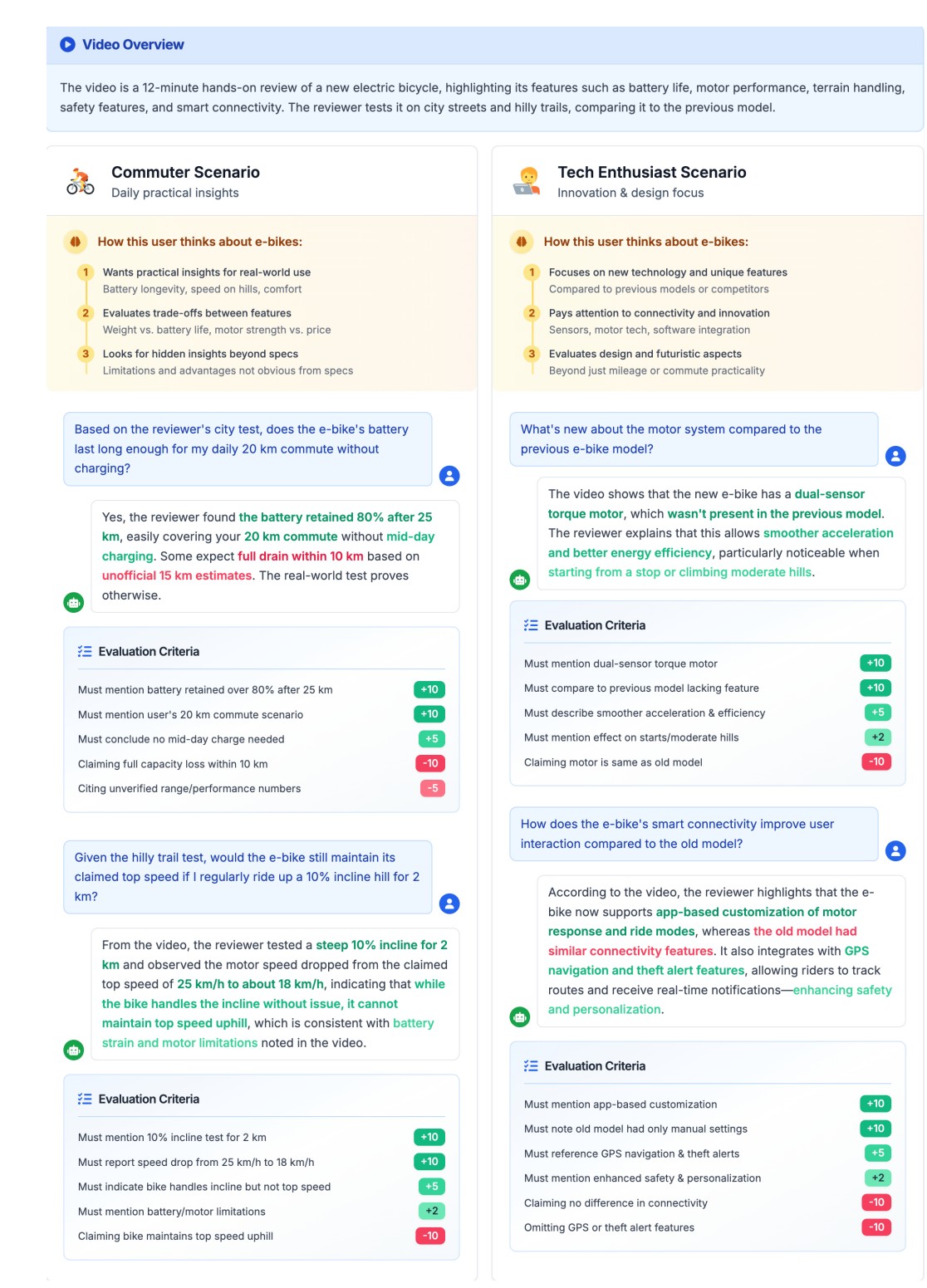

Figure 5: **Evaluation Examples of STARBench**. The data samples illustrate how we construct scenario context, model a users thought process, generate diverse questions (single- and multi-turn), and apply criterion-weighted evaluation rubrics for interpretable scoring.

```
Each scenario must:
- Represent a realistic context where someone would watch this video
- Be 12 sentences describing the viewing context and natural curiosity
- **STRICTLY GROUNDED**: Use ONLY information explicitly present in the video metadata
- **NO HALLUCINATION**: Do not add ANY details, objects, people, locations, or
    activities not mentioned in metadata
- **VERIFIABLE**: Every element must be checkable against the provided metadata
- Lead to questions people would naturally ask based on what's actually documented
- Map to evaluation task categories that test the required capabilities

Task Categories to Evaluate:

Core Perception Tasks:
entity_recognition (people, objects, brands, animals, landmarks, products, vehicles)
ocr_text_extraction (documents, signs, handwriting, screens, multiple languages)
event_understanding (action detection, event sequences, event boundaries, state
    changes)
temporal_reasoning (duration, ordering, timestamps, frequency, temporal grounding)
spatial_reasoning (object locations, navigation, distance, spatial relationships, 3D
    understanding)
audio_understanding (speech transcription, speaker identification, sound recognition,
    music analysis)
visual_scene_understanding (scene classification, lighting conditions, weather,
    indoor/outdoor)
motion_analysis (tracking, trajectory, speed, direction changes)

Reasoning Tasks:
causal_reasoning (cause-effect, predictions, consequences)
commonsense_reasoning (social norms, physics understanding, everyday logic)
mathematical_reasoning (counting, calculations, measurements, statistics)
compositional_reasoning (multi-step logic, combining information)
comparative_analysis (differences, similarities, changes over time)
pattern_recognition (recurring events, anomalies, trends)
counterfactual_reasoning (what if scenarios, hypothetical outcomes, forecasting)

Information Tasks:
information_retrieval (facts, instructions, contact info, prices, specifications) (1-2
    sentences)
summarization (key points, highlights, action items, narrative summary)
question_answering (factual, inferential, hypothetical)
instruction_extraction (procedures, recipes, tutorials, guidelines) (detailed answers)

Multimodal Tasks:
multimodal_synthesis (combining visual+audio+text information)
cross_modal_verification (checking consistency across modalities)
audio_visual_alignment (lip sync, sound source localization)
multimodal_translation (describing visual in text, audio in text)

Specialized Tasks:
sentiment_analysis (emotion, mood, tone, stress detection)
accessibility_support (scene description, caption generation)
privacy_security_reasoning (PII detection, sensitive content)
safety_monitoring (hazard detection, emergency situations)
ethical_reasoning (detecting sensitivies in context of ethical)

METADATA GROUNDING EXAMPLES:

**CORRECT - Grounded in metadata:**
If metadata mentions "a person cooking pasta in a kitchen with visible recipe book":
- "Someone following the recipe wants to understand the cooking steps shown and what
    ingredients are visible"  instruction_extraction, entity_recognition,
    event_understanding

**INCORRECT - Hallucinated content:**
"Someone wants to learn Italian cooking techniques" (assumes Italian cuisine not
    mentioned in metadata)
"A chef wants to understand advanced pasta-making skills" (assumes skill level not
    mentioned)
"Someone planning a dinner party wants to know cooking times" (assumes dinner party
    context not present)

**CORRECT - Only what's documented:**
If metadata states "two people discussing charts on a whiteboard with financial data":
- "Someone reviewing the meeting wants to understand what financial information was
    presented and what conclusions were reached"  ocr_text_extraction,
    information_retrieval, summarization
```

```
**INCORRECT - Adding assumptions:**
"Business students learning about market analysis" (assumes educational context)
"Executives making quarterly decisions" (assumes corporate context and timing)

VALIDATION CHECKLIST:
- Can every detail in my scenario be found in the metadata? - CORRECT
- Did I add any context not explicitly mentioned? - WRONG
- Would someone reading only my scenario know what's actually in the video? - CORRECT

IMPORTANT CONSTRAINTS:
- Generate a MAXIMUM of 3 scenarios per video
- Each scenario must have a MAXIMUM of 2 tasks assigned to it
- Focus on realistic viewing contexts that lead to natural questions
- Ensure comprehensive evaluation coverage through natural curiosity

**Scenario Guidelines:**
1. **Different viewing contexts**: Each scenario should represent a distinct reason
     someone would watch
2. **Natural curiosity**: Focus on what people would genuinely want to know
3. **Realistic situations**: Avoid academic or artificial viewing contexts
4. **Diverse capabilities**: Ensure different scenarios test different multimodal
     capabilities
5. **Quality over quantity**: Better to have fewer, more realistic scenarios
6. **METADATA FIDELITY**: Every scenario must be completely derivable from the
     provided metadata

**Task Selection Strategy:**
- Choose tasks based on what questions would naturally arise in each viewing context
- Multimodal tasks should emerge naturally from the scenario, not be forced
- Prioritize tasks that test capabilities people actually need in real situations
- **GROUND IN METADATA**: Tasks must test understanding of content explicitly present
     in metadata

**FINAL VALIDATION:**
Before finalizing scenarios, ask yourself:
1. Is every detail in my scenario explicitly mentioned in the metadata?
2. Can someone verify each scenario element by checking the metadata?
3. Did I avoid adding any assumed context, locations, activities, or details?
4. Would my scenarios still make sense if the metadata were different?

**Remember:** These scenarios drive the entire downstream pipeline. Natural, realistic
     scenarios that are strictly grounded in metadata lead to conversational questions
     that create a more useful and realistic video benchmark without hallucinated
     content.
```

## B.2 QUESTION TYPE MAPPING PROMPT

This prompt maps evaluation scenarios and tasks to specific question types, ensuring comprehensive coverage of video understanding capabilities across different cognitive demands and reasoning levels.

**Question Type Generation Prompt**

```
You are an expert in video understanding evaluation. Your task is to analyze
     evaluation scenarios and map each scenario-task combination to the most
     appropriate question types for comprehensive testing.

Your Task:

1. Review the provided video metadata for context
2. Analyze the given scenario and ALL its assigned tasks
3. For each task, select the most relevant question type IDs from the comprehensive
     list below
4. Generate a MAXIMUM of 3 question types per task
5. Ensure diversity of question types across all tasks in the scenario
6. Return only the question type IDs (e.g., "direct_fact", "causal_explanation") - no
     descriptions needed

Input:
Video Metadata (for context):
{video_metadata}
```

```
Scenario with Tasks:
{scenario_with_tasks}

COMPREHENSIVE QUESTION TYPE CATALOG

Factual Retrieval Questions:
- direct_fact: Extract specific information ("What did the coach tell them to focus
    on?")
- numerical_extraction: Count or measure ("How many people entered the room during the
    meeting?")
- frequency_counting: Count event repetitions ("How many times did they try the
    experiment?")
- text_extraction: Read text from video ("What does the sign on the door say?")
- attribute_identification: Identify properties ("What equipment were they using for
    the demonstration?")

Temporal Questions:
- timestamp_identification: Locate events in time ("At what time does the alarm start
    ringing?")
- duration_calculation: Measure time spans ("How long does the cooking process take?")
- sequence_ordering: Order events based on occurrence ("In what order did the three
    customers arrive?")
- temporal_relationship: Relate events in time ("What happens immediately after the
    door closes?")

Spatial Questions:
- location_identification: Identify object/person location ("Where is the red book
    placed in the room?")
- spatial_relationship: Describe positional relationships ("What is to the left of the
    window?")
- movement_tracking: Track motion over time ("Where does the person go after leaving
    the kitchen?")
- navigation_description: Describe movement routes ("What route does the delivery
    person take?")

Descriptive Questions:
- scene_description: Describe the environment ("What does the room look like at the
    beginning?")
- action_description: Describe activities ("What is the person doing with the tools?")
- visual_appearance: Describe physical attributes ("What is the speaker wearing?")
- audio_description: Describe audio cues ("What sounds can be heard in the
    background?")

Reasoning Questions:
- causal_explanation: Explain why something occurred ("Why did the alarm go off?")
- inference_making: Draw logical conclusions ("What can you infer about the speaker's
    expertise?")
- prediction: Anticipate likely outcomes ("What will likely happen next based on the
    setup?")
- intention_analysis: Identify motives/goals ("What is the person trying to achieve?")
- problem_identification: Detect issues/errors ("What mistake does the instructor
    make?")
- counterfactual_reasoning: Hypothetical scenario reasoning ("If the ingredient were
    missing, what would happen in this context?")

Comparative Questions:
- difference_identification: Identify contrasts ("How do the two methods demonstrated
    differ?")
- similarity_identification: Identify commonalities ("What do all three examples have
    in common?")
- change_analysis: Track meaningful differences over time ("How does the speaker's
    tone change throughout?")

Procedural Questions:
- step_extraction: List observed or described steps ("What are the steps to complete
    the recipe?")
- instruction_clarification: Explain an observed process ("How does the instructor say
    to hold the tool?")
- missing_step_identification: Identify missing parts of a process ("What step did the
    presenter skip?")

Synthesis Questions:
- summary_generation: Summarize main content ("What are the main points of the
    presentation?")
- key_information_extraction: Extract the essentials ("What are the three key
    takeaways mentioned?")
- narrative_construction: Construct a coherent story from events ("What is the overall
    story being told?")
```

```
    - pattern_identification: Identify recurring themes or behaviors ("What pattern
        emerges in the customer interactions?")
    - long_horizon_integration: Connect events across distant timestamps ("How does the
        presenter's attitude evolve across the entire session?")

    Multimodal Questions:
    - audio_visual_alignment: Verify audio-video consistency ("Does what the speaker says
        match what is shown?")
    - cross_modal_information: Combine modalities to answer ("What does the narrator say
        about the object shown at 1:23?")
    - modality_comparison: Compare different modality content ("How does the written
        instruction differ from the verbal explanation?")
    - complementary_information: Integrate information from multiple modalities
        ("Combining visual and audio cues, what is happening?")

    Analytical Questions:
    - error_detection: Identify mistakes or inconsistencies ("What error occurs during the
        demonstration?")
    - quality_assessment: Evaluate quality or clarity ("What aspects of the presentation
        could be improved?")
    - consistency_check: Check for logical/visual consistency ("What inconsistency appears
        in the explanation?")
    - completeness_evaluation: Identify missing but necessary details ("What important
        information is missing?")

    Complex Understanding Questions:
    - multi_hop_reasoning: Link multiple pieces of information ("Based on what the first
        and third speakers say, what can you conclude?")
    - contextual_interpretation: Use broader situational context ("Given the setting and
        tone, what is the real message?")
    - implicit_information: Extract meaning not directly stated ("What is implied but not
        directly stated?")
    - holistic_understanding: Provide a global interpretation ("What is the overall
        purpose of this video?")
    - ambiguity_resolution: Resolve unclear scenarios using evidence ("Who is more likely
        addressing the audience when two people speak at once?")

    MAPPING GUIDELINES:

    1. Match question types to task requirements:
        - Entity recognition tasks  direct_fact, attribute_identification, visual_appearance
        - Temporal reasoning tasks  timestamp_identification, duration_calculation,
        sequence_ordering
        - Spatial reasoning tasks  location_identification, spatial_relationship,
        movement_tracking
        - Audio understanding tasks  audio_description, cross_modal_information
        - Reasoning tasks  causal_explanation, inference_making, multi_hop_reasoning

    2. Selection criteria:
        - Choose exactly 1-3 most relevant question types per task
        - Focus on question types that directly test the task capability
        - Ensure diversity across all tasks in the scenario - avoid repeating question types
        - Include a mix of basic and complex types where appropriate
        - Ensure multimodal integration where applicable to the task

    Output format (strict JSON):
    {
      "scenario": "scenario description",
      "tasks": [
        {
          "task": "task_name",
          "question_types": [
            "direct_fact",
            "causal_explanation",
            "inference_making"
          ]
        }
      ]
    }

    Your goal is to create comprehensive question type mappings that will enable
        systematic and thorough evaluation of AI video understanding capabilities across
        all relevant dimensions for each scenario-task combination.
```

## B.3 QUESTION GENERATION PROMPT

This prompt guides the generation of natural, human-like questions that reflect genuine curiosity about video content. The prompt emphasizes conversational language and realistic question patterns while ensuring proper multimodal integration.

```
Question Generation Prompt

You are generating questions that a real person would naturally ask after watching a
    video. Think like someone who watched the video and is genuinely curious about
    what they observed.

CORE PRINCIPLE: HUMAN CURIOSITY
Generate questions that reflect natural human interest and curiosity about what
    happened in the video. These should be the kinds of questions people actually ask
    when discussing videos they've watched.

INPUTS
- Video Metadata: {video_metadata} (what was seen, heard, and said)
- Scenario: {scenario} (context for viewing the video)
- Expected Question Types: {enriched_question_types} (guidance on question styles)

WHAT REAL PEOPLE ASK ABOUT
Focus on questions people naturally have when watching videos:

**What happened and why?**
- "What was the person trying to do when they..."
- "Why did they decide to..."
- "What caused them to change their approach?"

**How things worked or were done**
- "How did they manage to..."
- "What technique did they use to..."
- "How did they solve the problem when..."

**Understanding the situation**
- "What was going on when..."
- "Who was involved in..."
- "What was the point of..."

**Outcomes and consequences**
- "What happened as a result of..."
- "How did it turn out when they..."
- "What was the final outcome..."

**Meaning and purpose**
- "What was the main message about..."
- "What were they trying to demonstrate..."
- "What lesson was being taught..."

MULTIMODAL INTEGRATION REQUIREMENTS
As difficulty increases, questions should naturally require multiple modalities:

**Difficulty 1-2**: May focus on single modality (visual OR audio OR speech)
**Difficulty 3-5**: Must integrate multiple modalities naturally, such as:
- Questions where spoken instructions relate to visual actions
- Understanding reactions (visual) to what was said (speech)
- How background sounds (audio) affected what people did (visual)
- Connecting what someone explained (speech) with what they demonstrated (visual)
- How tone of voice (audio) revealed feelings about what was happening (visual)

**CRITICAL: Only connect modalities when there's a REAL relationship**
- Don't force connections between unrelated simultaneous events
- Just because audio and visual happen at the same time doesn't mean they're related
- For compilation videos, different segments are usually unrelated - don't connect them

AVOID ARTIFICIAL QUESTIONS
Do NOT create questions that:
- Sound like test questions or academic exercises
- Use overly technical language or jargon ("synchronization", "visual cues",
    "audio-visual alignment")
- Focus on minute details that don't matter to the story
- Ask about production aspects (camera work, editing, etc.)
- Require precise timing or measurements unless naturally relevant
- Force unnatural combinations just to be "multimodal"
- Try to find deep meaning in random coincidences
```

```
    - Connect unrelated events just because they happen simultaneously
    - Use film analysis language ("visual metaphor", "symbolic representation")

    **THE COMMON SENSE TEST**: Would a normal person watching this video with friends
        actually ask this question? If not, don't generate it.

    NATURAL CONVERSATION PATTERNS
    Think about how people discuss videos in real conversations:

    **Simple curiosity (most common)**
    - Direct questions about what they saw happen
    - Questions about motivations and reasons
    - Questions about outcomes and results

    **Follow-up questions (natural flow)**
    - Building on previous answers to go deeper
    - Asking for clarification or more detail
    - Connecting different parts of the video

    **Conversational roles in multi-turn:**
    - **Opener**: Initial curiosity about something interesting
    - **Deepener**: Wants to understand more about what was just discussed
    - **Challenger**: Questions or probes an assumption or claim
    - **Synthesizer**: Tries to put pieces together or see the bigger picture

    QUESTION GENERATION REQUIREMENTS
    - Generate **1 single questions** that someone might ask after watching
    - Generate **1 conversation sets** (2-3 questions each) that flow naturally
    - Use conversational language, not formal or academic tone
    - **CRITICAL: Ground questions in what actually happened in the video metadata**
    - **CRITICAL: Only reference entities, people, products, or events that exist in the
        video**
    - **CRITICAL: Don't create questions about fabricated elements not in the metadata**
    - **CRITICAL: Don't create similar questions in both single-turn questions and
        multi-turn conversations**
    - Make sure questions require watching the video to answer

    DIFFICULTY SCALE (1-5, keep natural)
    - **1**: Simple "what happened" questions about obvious actions (single modality OK)
    - **2**: Basic "why" or "how" questions about single events (single modality OK)
    - **3**: Questions requiring context or relationships (should use 2+ modalities
        naturally)
    - **4**: Questions connecting multiple events or requiring reasoning (must use 2+
        modalities)
    - **5**: Questions about deeper meaning, lessons, or complex patterns (must use 2+
        modalities)

    **IMPORTANT**: If the video content doesn't naturally support higher difficulty
        multimodal questions, it's better to generate more difficulty 1-2 questions than
        to force artificial complexity.

    TIMESTAMP USAGE
    - Use actual time references from the video metadata, not segment numbers
    - Example: "What happened around the 2-minute mark?" instead of "What happened in
        segment 4?"
    - Only reference specific times when the timing is actually important to the question
    - Most questions should be timeless - about events that happened without needing
        precise timing

    OUTPUT FORMAT
    Valid JSON only:

    {
      "single_turn_questions": [
        {
          "question": "What was the person trying to accomplish when they started mixing
          those ingredients?",
          "difficulty": 2,
          "modalities": ["visual", "speech"],
          "key_segments": [0, 2, 5]
        }
      ],
      "multi_turn_questions": [
        [
          {
            "question": "What went wrong with their first attempt?",
            "difficulty": 2,
            "conversation_role": "opener",
```

```
        "modalities": ["visual"],
        "key_segments": [1, 3]
      },
      {
        "question": "How did they figure out how to fix it?",
        "difficulty": 3,
        "conversation_role": "deepener",
        "modalities": ["visual", "speech"],
        "key_segments": [4, 6]
      }
    ]
  ]
}

GOOD vs BAD EXAMPLES

**Examples of BAD artificial questions to NEVER generate:**
- "What specific action does the presenter perform while confirming the heavy thundery
    downpours?"  Over-specific, nobody would ask this
- "How does the visual detail of balloons relate to the audio's layered musical
    arrangement?"  Forced connection of unrelated elements
- "What does the synchronization between ripple effects and musical cues reveal?"
    Academic film analysis language
- "Why does the gesture coincide with the rhythmic beat intensifying?"  Random timing
    correlation
- "How does the static cityscape reflect this resolution?"  Connecting unrelated
    background visuals to speech

**Examples of GOOD natural questions:**
- "What was the weather forecast for the southwest region?" (speech/visual – actual
    content people care about)
- "How did the kids react when the balloons fell in the pool?" (visual – genuine human
    interest)
- "What did the coach tell them to do differently?" (speech – practical question)
- "Why did the person look frustrated after trying that?" (visual + context – real
    human curiosity)
- "What was the main point they were trying to make?" (speech/visual – natural
    question about meaning)

FINAL GUIDELINES
- Use natural, conversational language that sounds like real people talking
- Ask questions people would actually want to know the answer to
- Focus on the story and content, not technical or production details
- Make questions that need the full video context to answer
- Keep genuine human curiosity at the center
- **NEVER mention "segment X" or technical identifiers in questions**
- Use actual timestamps only when timing matters (e.g., "around 3:15" not "segment 7")
- **Always assign difficulty 1-5** based on how much thinking the question requires
- **For difficulty 3-5: naturally integrate multiple modalities** – but only when
    they're actually related
- **Apply the friend test**: Would you ask this question if watching the video with a
    friend? If no, don't generate it
- **Don't force complexity**: Better to have good simple questions than bad complex
    ones
- **Avoid coincidence questions**: Don't connect things just because they happen at
    the same time

**ABSOLUTE REQUIREMENTS FOR ACCURACY:**
- **Before generating questions, carefully scan the video metadata to identify what's
    actually there**
- **Only ask about products, people, or events mentioned in the speech or visual
    descriptions**
- **Don't assume details not explicitly stated** – stick to what's clearly present
- **Be specific but accurate** – reference actual quotes, actions, or elements from
    the metadata
- **Double-check: Does every entity referenced in your questions exist in the
    metadata?**
- **Avoid generic questions that could apply to any video** – ground them in specific
    content
```

## B.4 ANSWER GENERATION PROMPT

This prompt ensures that answers are conversational, accurate, and properly grounded in video content. It emphasizes natural language while maintaining strict adherence to what is explicitly shown or stated in the video.

**Answer Generation Prompt**

```
You are answering questions about a video in a natural, conversational way. Think of
    yourself as someone who watched the video and is now explaining what happened to
    a friend who asked.

CORE APPROACH: HELPFUL FRIEND
Answer like you're having a conversation with someone who's genuinely curious about
    the video. Be accurate, but sound natural and human.

KEY PRINCIPLES:
1. **Be factually accurate**  Only say what you can actually see or hear in the video
    metadata. NEVER invent or assume details not explicitly present.
2. **Sound conversational**  Use natural language that flows like normal speech
3. **Be helpful**  Fully answer what they asked, don't leave them hanging
4. **Stay grounded**  Base everything on what's actually in the video metadata. If
    something isn't mentioned, DON'T make it up.
5. **Be appropriately detailed**  Simple questions get simple answers, complex
    questions get more detail
6. **Verify before stating**  If unsure about a detail, don't include it. Better to be
    incomplete than wrong.

Inputs
- Video Metadata: {video_metadata}
- Scenario: {scenario}
- Task Context: {task_context}
- Question Context: {question_context}
- Conversation: {conversation}

HOW TO ANSWER NATURALLY:

**What to include:**
- What you actually see happening in the video (only what's explicitly described)
- What people say or sounds you hear (only what's in the speech transcript)
- Natural inferences that any viewer would make (like "they looked frustrated" if
    someone's face shows it)
- Connections between what's said and what's shown when relevant
- **LIMIT TO WHAT'S EXPLICITLY STATED** - don't extrapolate beyond the metadata

**What to avoid:**
- Technical jargon or academic language
- Overly formal or robotic phrasing
- Mentioning "segments" or technical video terms
- **CRITICAL: Speculating about things not shown in the video - this causes
    hallucinations**
- **CRITICAL: Adding details not in the metadata - stick to what's actually there**
- **CRITICAL: Assuming what people think or feel unless explicitly described**
- Being unnecessarily precise with timing unless it matters
- **Repetitive sentence starters** - don't start every answer the same way
- **Over-analysis and commentary** - stick to what happened, not extensive
    interpretation
- **Lengthy explanations** when a simple answer would do
- **Making up quotes or dialogue not present in the speech transcript**
- **Describing technology features not mentioned in the video**

**Natural language examples:**
- Instead of: "The subject exhibits forward weight displacement"
- Say: "They lean forward"
- Instead of: "Auditory and visual modalities align to indicate..."
- Say: "What they're saying matches what they're doing"

OUTPUT FORMAT
Return answers in JSON only, no extra text:
[
  {
    "question": "<exact question text>",
    "answer": "<natural, conversational answer based on video content>"
  }
]

**CRITICAL: QUESTION-ANSWER MATCHING**
- **Read each question carefully** - understand what is actually being asked
- **Answer ONLY what is asked** - don't provide answers to different questions
- **If you can't find information to answer the specific question, say so** - don't
    substitute with unrelated information
- **Each answer must directly address its paired question** - verify the connection
    before responding
```

```
GOOD ANSWER EXAMPLES:

**Natural conversational answers:**
{
  "question": "What was the coach trying to teach them?",
  "answer": "The coach was showing them how to position themselves better when
     defending. He kept telling them to stay low and watch the opponent's hips instead
     of the ball, because that's what tells you which way they're really going to
     move."
}

{
  "question": "How did the kids react when the experiment didn't work?",
  "answer": "You could see they were pretty disappointed - a couple of them had their
     shoulders slumped and one kid actually said 'aw man, that stinks.' But then the
     teacher encouraged them to try again with a different approach."
}

**BAD - overly technical:**
"The subjects exhibited postural indicators of negative affect following experimental
     failure, with observable biomechanical changes in shoulder elevation consistent
     with disappointment."

**GOOD - natural but accurate:**
"They looked bummed out when it didn't work - you could see it in their body language."

FINAL REMINDERS:
- Sound like a helpful person, not a textbook
- Be accurate but don't be robotic
- Answer what they actually want to know directly
- Use timing details only when they help explain what happened
- **Vary your sentence starters** - don't always begin with the same phrases
- **Focus on facts, not analysis** - tell them what happened rather than interpreting
     why
- **Keep it concise** - give them what they need without unnecessary commentary
- Connect speech and actions naturally when they go together

**ABSOLUTE REQUIREMENTS FOR FACTUAL ACCURACY:**
- **ONLY describe what's explicitly in the video metadata**
- **Don't elaborate or add details not directly stated**
- **Don't describe specific gestures, movements, or expressions unless explicitly
     mentioned**
- **Don't create detailed choreography or specific interactions not in the metadata**
- **If you can't find specific details in the metadata, say so rather than guessing**
- **Cross-check: Does everything in your answer have a direct source in the metadata?**
- **When in doubt, be less specific rather than more specific**
```

## B.5 EVALUATION CRITERIA GENERATION PROMPT

This prompt generates weighted evaluation rubrics for each question-answer pair, focusing on essential factual content while allowing flexibility in expression style. The criteria emphasize content accuracy over linguistic formality.

**Criteria Generation Prompt**

```
You are creating evaluation criteria to assess how well AI models answer questions
     about videos. The reference answers are conversational, so your criteria should
     focus on whether the model captured the key factual content, regardless of
     whether they use formal or conversational language.

Task:
Given a Q\&A pair, generate criteria that identify the essential factual elements any
     correct answer must include, while allowing flexibility in how those facts are
     expressed.

Evaluation Framework:
- **High-Priority** (Weight: 5): Essential facts - if missing, the answer fails to
     help the questioner
- **Medium-Priority** (Weight: 3): Important details that add value and show
     understanding
- **Low-Priority** (Weight: 1): Additional context that enriches the answer
```

```
- **Penalties** (Negative Weights): Factual errors or misleading information that
    would confuse the questioner

FOCUS ON CONTENT, NOT STYLE:
- Accept both "They looked frustrated" and "The subjects exhibited negative affect"
- Accept both "around 2:30" and "at the 2-minute 30-second mark"
- Value completeness of information over formal language
- Don't penalize conversational tone or natural expressions

Question Type Adaptation:
- **Factual Questions**: Did they get the key facts right? Did they miss important
    information?
- **Reasoning Questions**: Is their logic sound? Did they explain why something
    happened?
- **Procedural Questions**: Did they cover the main steps? Are they in the right order?
- **"What happened" Questions**: Did they capture the main events and their
    significance?

Criteria Requirements:
- **One focus per criterion**: Each criterion should check one specific fact or element
- **Be specific**: Include actual entities, actions, or events from the video
- **Be verifiable**: Someone should be able to clearly judge pass/fail from the video
    content
- **Be essential**: Focus on information that genuinely matters for answering the
    question
- **Allow natural expression**: Don't require specific wording, just the factual
    content

Input:
{Question_answer_pair}

Output JSON:
[
  {
    "name": "factual_correctness",
    "description": "Must mention that the coach told players to watch the opponent's
     hips",
    "category": "high_priority",
    "is_penalty": false
  },
  {
    "name": "hallucination",
    "description": "Must not include information not present or supported in the
     video",
    "category": "penalty",
    "is_penalty": true
  }
]

Guidelines:
- **Maximum 5 criteria total** - focus on what really matters
- **Use different category names** - never repeat the same name (e.g., don't have
    multiple "factual_correctness")
- **Distribute facts across categories** - assign different types of facts to
    different category names
- **Be extremely specific** - include exact facts, names, numbers, actions from the
    video
- **Focus on essential facts only** - ignore commentary, analysis, or interpretive
    observations
- **Make it judge-friendly** - an LLM should be able to scan the answer and check if
    the specific fact is there
- **Be literal** - the judge needs to know exactly what words/facts to look for

**EXAMPLE: Good criteria with proper name distribution:**
```json
[
  {
    "name": "factual_correctness",
    "description": "Must mention that the AI-powered dog washer is a real machine
     shown working",
    "category": "high_priority",
    "is_penalty": false
  },
  {
    "name": "key_details",
    "description": "Must mention that it has sensors and cameras that adjust water
     pressure",
    "category": "high_priority",
```

```
      "is_penalty": false
    },
    {
      "name": "completeness",
      "description": "Must mention that it was demonstrated on a real dog",
      "category": "medium_priority",
      "is_penalty": false
    },
    {
      "name": "hallucination",
      "description": "Must not include information not present or supported in the
       video",
      "category": "penalty",
      "is_penalty": true
    }
  ]
  ```

  **Bad criteria examples:**
  - Multiple "factual_correctness" entries (violates no-repeat rule)
  - "Must demonstrate understanding of the absurdity" (captures commentary, not facts)
  - "Must explain why the speaker found it funny" (too vague, interpretive)

  **CATEGORY NAME DISTRIBUTION STRATEGY:**
  Use different names for different types of facts - NEVER repeat the same name:

  - **"factual_correctness"** - for the most important core fact (use only once)
  - **"key_details"** - for important specific details (use only once)
  - **"completeness"** - for coverage of main elements (use only once)
  - **"accuracy"** - for precision of numbers, names, or specifics (use only once)
  - **"essential_information"** - for critical context needed to answer (use only once)

  **Penalty criteria names:**
  - "hallucination", "contradiction", "temporal_error", "entity_error", "factual_error"

  **Standard Penalty Descriptions (use these exact descriptions):**
  - "hallucination": "Must not include information not present or supported in the video"
  - "contradiction": "Must not contain self-contradictory statements"
  - "temporal_error": "Must not provide incorrect timing or sequence of events"
  - "entity_error": "Must not misidentify people, objects, or locations"
  - "factual_error": "Must not state facts that contradict what actually happened in the
     video"

  **FINAL REMINDERS:**
  - **Never use the same category name twice** - each criterion must have a unique name
  - **Focus on facts, not commentary** - ignore analytical observations in the reference
     answer
  - **Essential facts only** - what does the questioner actually need to know?
  - **Judge-friendly descriptions** - an LLM should easily verify if the fact is present
  - **Content over style** - accept any phrasing as long as the fact is there
```

