# OpenReview forum: "Towards Multimodal Understanding, Reasoning, and Tool Usage across Vision, Speech, and Audio in Long Videos"
_ICLR.cc/2026/Conference — ICLR 2026 Conference Withdrawn Submission_

### Official Review · Reviewer_cjgM · 2025-10-31

**Soundness:** 3
**Presentation:** 3
**Contribution:** 2
**Rating:** 4
**Confidence:** 4

**Summary:**

This paper presents STARBench, a benchmark designed to evaluate the multimodal understanding capabilities of large language models (MLLMs) in long-form videos, integrating vision, speech, and audio signals. It also introduces STARAgent, an agentic system for analyzing long videos using a combination of preprocessing, search, and refinement tools. The authors highlight the limitations of current state-of-the-art models and demonstrate how STARBench can be used to systematically evaluate and improve multimodal reasoning over extended temporal contexts.

**Strengths:**

+ STARBench offers a novel and comprehensive framework for evaluating multimodal understanding in long-form videos. Unlike prior benchmarks, it integrates audio, speech, and visual signals, and focuses on tasks involving reasoning, tool usage, and cross-modal integration.
+ The benchmark uses a human-validated pipeline for creating and scoring the questions, which adds credibility to the evaluation process and ensures that the tasks are relevant and grounded in real-world video data.

**Weaknesses:**

- I acknowledge the contribution of this paper in benchmark construction; however, the evaluation design appears inadequate in several aspects: (1) Regarding closed-source multimodal large models, only Gemini was included for comparison, while other models such as GPT-4o were omitted. These models could help reveal distinctive characteristics of the proposed dataset. (2) There is a lack of comparison with video-agent-related methods. Among all the compared approaches, only those based on multimodal large models were considered, while only the method proposed in this study incorporates multiple tools, which clearly constitutes an unfair comparison.
- Some experimental findings warrant further in-depth analysis: (1) Why does the performance of the proposed method remain similar across videos of varying durations? (2) In Table 5, why does the 'S' modality alone achieve the best performance (even the combination of S+V performs worse than S alone, and further incorporating the A modality leads to additional degradation)? Does this suggest that the visual modality contributes minimally in this dataset? (3) Why are the experimental results for single-turn and multi-turn questions so similar? Theoretically, multi-turn questioning represents a more challenging setting. Does this indicate potential biases in the annotation of the benchmark?
- Both the data annotation process and the agent model constructed in this study utilize the QWen model, which appears methodologically questionable.

**Questions:**

Please refer to the weakness part.

---

### Official Review · Reviewer_WiQH · 2025-11-01

**Soundness:** 2
**Presentation:** 2
**Contribution:** 2
**Rating:** 4
**Confidence:** 5

**Summary:**

The paper introduces STARBench, a benchmark for long-form, multimodal video understanding, addressing gaps in existing benchmarks. STARBench uses open-ended, intent-driven questions that support dialogues and tool use across video, audio, and speech contexts. And the proposed STARAgent achieves 44.66% accuracy through structured reasoning on complex videos.

**Strengths:**

- The benchmark spans multiple dimensions, including reasoning tasks and multimodal tasks, demonstrating its wide applicability and rigorous challenges.

**Weaknesses:**

- The paper exclusively uses the Qwen3-30B-A3B model to generate answers, which may introduce biases in the generated outputs.

- The evaluation employs only the Qwen3-14B model as the benchmark, which is also likely to lead to biased evaluation results due to its limitations.
- The range of models tested in this study is insufficiently broad. The paper should consider testing more advanced models, such as Gemini-2.5-pro and GPT-5, and enabling the thinking mode of these models to fully evaluate their potential.

- The experiments are limited to relatively small models with around 7B size. The lack of larger and more powerful models in the evaluation limits the comprehensiveness of the results.

**Questions:**

- Why did the Qwen-Omni fail to show advantages in the audio-visual alignment task?
- How does StarAgent perform on the DailyOmni, WorldSense, and VideoHolmes benchmarks?

---

### Official Review · Reviewer_orZe · 2025-11-01

**Soundness:** 2
**Presentation:** 1
**Contribution:** 2
**Rating:** 4
**Confidence:** 5

**Summary:**

This paper presents **STARBench**, a benchmark for **long-form multimodal video understanding** that integrates vision, speech, and ambient audio. It emphasizes **open-ended, intent-driven questions** and **graded evaluation rubrics**, enabling interpretable assessment beyond multiple-choice formats. The authors also introduce **STARAgent**, an agentic system combining preprocessing, search, and reasoning tools for long-video analysis. Evaluations show the task’s difficulty—**Gemini-2.5-Flash** achieves **52.95%**, open-source models <25%, and **STARAgent** 44.66%. The benchmark and pipeline are **scalable, human-validated, and reproducible**, providing a foundation for future research.

**Strengths:**

- Well-motivated benchmark: Addresses a key gap by combining long-term reasoning and multimodal understanding.
- Interpretable evaluation: Uses open-ended questions and graded rubrics for finer diagnostic insights.
- Agentic complement: STARAgent demonstrates structured reasoning** for long-video comprehension.
- Scalable and reproducible: The human-validated pipeline ensures quality and broad applicability.
- Public release: Code, dataset, and pipeline enhance research transparency and community impact.

**Weaknesses:**

- **Lack of comparison with domain benchmarks:**
  While STARBench is well-motivated, it **does not include comparisons with influential benchmarks** in *audio-visual understanding* [1–2] or *video reasoning* [3–5]. Such comparisons would help position STARBench in terms of **coverage, difficulty, and evaluation philosophy** relative to established efforts.

- **Limited experimental comparisons and insufficient baseline coverage:**
  1. **Absence of multi-agent baselines:** STARAgent is not compared with related *multi-agent* systems such as **Daily-Omni** [2], limiting insight into its improvements over existing frameworks.
  2. **Need for cross-benchmark evaluation:** Evaluating STARAgent on additional long-form multimodal benchmarks could better demonstrate the **generality and scalability** of the multi-agent framework.
  3. **Insufficient model and method baselines:** As a benchmark paper, STARBench would benefit from **more diverse baselines** (both closed- and open-source) to establish **stronger diagnostic references**.

- **Incomplete experimental details:**
  The paper lacks crucial implementation details—such as **number of frames or clips per video**, **handling of videos exceeding 30s**, and **treatment of truncated or dropped audio** for models with limited context capacity. These details are vital to **validate experimental soundness and reproducibility**.
---
Reference

[1] WorldSense: Evaluating Real-world Omnimodal Understanding for Multimodal LLMs,https://arxiv.org/abs/2502.04326

[2] Daily-Omni: Towards Audio-Visual Reasoning with Temporal Alignment across Modalities,https://arxiv.org/abs/2505.17862

[3] RTV-Bench: Benchmarking MLLM Continuous Perception, Understanding and Reasoning through Real-Time Video, https://arxiv.org/abs/2505.02064

[4] Video-MMMU: Evaluating Knowledge Acquisition from Multi-Discipline Professional Videos

[5] Video-Holmes: Can MLLM Think Like Holmes for Complex Video Reasoning? https://arxiv.org/abs/2505.21374

**Questions:**

The key concerns are outlined in the weaknesses section.

---

### Official Review · Reviewer_bBJM · 2025-11-05

**Soundness:** 1
**Presentation:** 3
**Contribution:** 2
**Rating:** 4
**Confidence:** 4

**Summary:**

This work proposes STARBench, a diagnostic benchmark designed for long-form, multimodal video understanding. STARBench features open-ended, intent-driven questions that reflect how humans naturally engage with video content. It supports single- and multi-turn dialogues, encompassing multimodal reasoning and agentic tool-use tasks across rich video, audio, and speech contexts.

**Strengths:**

1. The proposed benchmark supports single- and multi-turn dialogues, encompassing multimodal reasoning and agentic tool-use
tasks across rich video, audio, and speech contexts.
2. The proposed STARAgent outperforms the selected MLLMs.

**Weaknesses:**

1. SVBench should be included in the dataset comparison since it contains multi-turn Q&As.
2. The comparison of video duration and numbers with existing benchmarks is not provided.
3. Important details about the evaluation are unknown, including video sampling, devices, and maximum context length, which can significantly impact the evaluation results.
4. The authors only evaluate MLLMs of the 7B/8B size, while the parameter scaling law on this benchmark is unclear.
5. The generalized performance of the proposed STARAgent on other video benchmarks is unknown.
6. Only one closed-source MLLM is evaluated, which weakens the results of this paper.
7. The rationality of the agentic tasks is insufficiently analyzed. More analysis about the tool design, task requirement of tools, and protocol of tool calling should be done.
8. As the proposed STARAgent utilizes multiple large models and its total parameters are far more than those of baselines, the performance comparison is unfair.

Minor: There are so many benchmarks named StarBench. Please consider renaming your benchmark.


References:
[1] Yang, Z., Hu, Y., Du, Z., Xue, D., Qian, S., Wu, J., ... & Xu, C. (2025). Svbench: A benchmark with temporal multi-turn dialogues for streaming video understanding. ICLR.

**Questions:**

Please reply to Weaknesses.

---

### Note · Authors · 2025-11-13

I have read and agree with the venue's withdrawal policy on behalf of myself and my co-authors.